# Effects of Dietary Lipid Levels on Growth and Gonad Development of *Onychostoma macrolepis* Broodfish

**Jishu Zhou** [1,*]**, Peng Feng** [1]**, Yang Li** [1]**, Hong Ji** [1] **and Enric Gisbert** [1,2]

1. College of Animal Science and Technology, Northwest A & F University, Yangling 712100, China
2. Aquaculture Program, Institut de Recerca i Tecnologies Agroalimentaries (IRTA), Centre de Sant Carles de la Rapita, Crta. de Poble Nou km 5.5, P.O. Box 200, 43540 Tarragona, Spain
* Correspondence: zhoujishu@163.com or jishuzhou@nwafu.edu.cn

**Abstract:** To assess the lipid requirements of O. macrolepis broodstock, five iso-nitrogenous diets (39 g kg$^{-1}$) with five lipid levels, 50 (5 L), 70 (7 L), 90 (9 L),110 (11 L), and 130 (13 L; g kg$^{-1}$), were made. A total of 105 three-year-old individuals (50.11 ± 2.86 g per fish) were divided into five groups (triplicate per group) and were fed with the diets, respectively, for eight weeks. Then, the fish were sampled, and items were determined. The results showed that growth rate and feed efficiency ratio were not significantly affected by diets ($p > 0.05$). A clear dose–response effect of dietary lipid was observed on somatic indexes of gonad indexes of the O. macrolepis brookstock, with the highest values corresponding to fish fed 9 and 11 g kg$^{-1}$ lipids, in contrast, gonad indexes were reduced as dietary lipid moved away from this level. The other somatic indexes, such as viscerosomatic index, perivisceral fat index, etc., were not significantly affected by diets ($p > 0.05$). The content of crude lipid and crude protein in carcass, hepatopancreas, and gonad were not significantly affected by dietary lipid levels ($p > 0.05$). The gonad fatty acids of 16:0 and 22:6n-3 decreased and 18:2n-6 increased with the increasing lipid level, being significantly altered by diets ($p < 0.05$). The histological features of the gonad showed no significant difference among the five diets ($p > 0.05$). The relative expression of sex steroid-synthesizing proteins (fshr, 3β-hsd, 17β-hsd, aro., and star.) in the gonad of fish was most significantly highly expressed in the 9 L and 11 L groups ($p < 0.05$). The results suggested that a proper dietary lipid level of 90–110 g kg$^{-1}$ could maintain gonad development of O. macrolepis broodstock without affecting growth performance.

**Keywords:** aquaculture; lipid; broodstock; growth; gonad development





## 1. Introduction

Given the increasing importance of domestication in aquaculture, there is a need for an increased focus on the role of nutrition in broodstock and gonad maturation, as well as in improving larval quality and performance. Izquierdo et al. [1] pointed out that broodstock nutrition of fin fish nutrition was one of the most poorly understood and researched areas almost two decades ago, and significant advances have been achieved in this area [2] in these years; however, there are still important gaps of knowledge related to fish broodstock nutrition. Considering the broad range of species and families important to aquaculture, and the relative scarcity of research into broodstock nutrition, it is difficult to generalize the nutritional requirements of broodfish. Some sporadic research reported the effect of vitamins [3–6] and protein and carbohydrate levels [7] on the reproductive performance of some broodstock fish. Lipids play an important role as sources of metabolic energy for somatic growth, and they are also the source of essential fatty acids required for the formation of cell membranes, as well as for gonad development and maturation [8–11]. In this context, the lipid and fatty acid composition of broodstock diets have been identified as major dietary factors that determine successful reproduction and survival of the offspring [1]. In this sense, several studies have shown that higher dietary lipid levels promoted gonad development, resulting in higher fecundity values, larger oocyte diameters,

and a greater number of mature oocytes [12–16]. In addition, dietary lipid levels have also been correlated to fish health, immunity, and condition [17], thus revealing the pleiotropic effects of this group of macronutrients on the organism.

The Onychostoma macrolepis (Bleeker, 1871), also named Gymnostomus macrolepis [18], is a freshwater benthopelagic cave fish species belonging to the Cyprinidae family that is distributed in the North of China, including the Jialing, Huai, Wei, Hai, and Yellow rivers [19]. This fish is very delicious and nutritious, being rich in protein and the important highly unsaturated fatty acids EPA (eicosapentaenoic acid) and DHA (docosahexaenoic acid), and is very much in demand, resulting in the high price of 200 RMB per kg in the local aquatic market [20]. Although the IUCN classify it as a species of least concern [21], this species has been listed as a National Second-class Protected Animal by the Chinese authorities (List of Aquatic Wild Animal Protection in China). The main reasons for such classification are anthropogenic threats [22] and overfishing and habitat (caves) degradation [23]. During these last years, the efforts for the conservation of this cave fish have increased, and several studies have been published regarding its protection [18], reproductive biology [24,25], and dietary protein requirements [26]. However, up to date, no available data exists on the dietary lipid requirements for this species; information that is critical for the successful artificial propagation of this species.

The objective of the present study was to investigate the optimal dietary lipid levels for the cavefish O. macrolepis broodstock. This assessment was conducted considering several parameters related to the somatic growth, gonad development, and health conditions of broodfish.

## 2. Materials and Methods

All experimental procedures were conducted in accordance with the Guidelines for Experimental Animals by the Animal Care and Use Committee of Northwest A & F University, China.

### 2.1. Experimental Diets, Fish and Feeding Procedures

Five isonitrogenous experimental diets (390 g kg$^{-1}$ protein) with increasing lipid levels (50, 70, 90, 110, and 130 g kg$^{-1}$) were formulated for O. macrolepis broodfish. Different dietary lipid levels were obtained by increasing soybean oil as the only source of dietary lipids (Table 1). The ingredients and the proximate composition of the experimental diets are presented in Table 1. The diets were named 5 L, 7 L, 9 L, 11 L, and 13 L, according to their percentage of lipid content (5, 7, 9, 11, and 13%, respectively). Regarding diet preparation, all ingredients were weighed and mixed for 15 min, after which, distilled water and oil were added and mixed. Once the desired consistency of the dough was reached, the mixture was then mechanically extruded to obtain suitable-sized pellets (1 mm). The pellets were dried in a convection oven at 25 °C for 24 h, stored in re-sealable plastic bags, and stored at −20 °C until their use.

A total of 105 healthy three-year-old O. macrolepis (body weight, BW = 50.1 ± 2.9 g, mean ± standard deviation) were obtained from Tian Yuan Shen Tai Aquaculture Farming (Ankang, Zheng Ping, Shaanxi, China) and transferred to the Ankang Fisheries Experimental and Demonstration Station of Northwest A&F University (Ankang, Shaanxi, China). Then, fish were randomly distributed regardless of their sex (no external sexual differentiation) into 15 rearing tanks of 130 L of volume (three replicates per experimental condition) and maintained in a recirculation aquaculture system (RAS) that guaranteed water quality utilizing ultraviolet, biological, and mechanical filtration. Tanks were equipped with aeration 24 h a day to maintain dissolved oxygen levels at 7.7 ± 0.2 mg L$^{-1}$ (mean ± standard deviation). The water temperature was 20.0 ± 1.5 °C, and the pH was 7.8 ± 0.5. The photoperiod was 12 h light and 12 h darkness. During the experimental period (8 weeks), fish were fed by hand to apparent satiation levels, three times a day (8:30, 12:30, and 16:30 h).

**Table 1.** Ingredients (g kg$^{-1}$, dry weight) and proximate composition (air dry basis, %) of experimental diets differ in their total crude lipids.

| Ingredients (g kg$^{-1}$) | Experimental Diets | | | | |
|---|---|---|---|---|---|
| | 5 L | 7 L | 9 L | 11 L | 13 L |
| Fish meal | 285 | 285 | 285 | 285 | 285 |
| Soybean meal | 145 | 141 | 140 | 140 | 140 |
| Rapeseed cake | 10 | 12 | 14 | 14 | 16 |
| Cottonseed meal | 195 | 214 | 230 | 247 | 263 |
| Wheat flour | 220 | 183 | 149 | 117 | 73 |
| Rice bran | 91 | 90 | 85 | 78 | 84 |
| Soybean oil | 4 | 25 | 47 | 69 | 89 |
| $\alpha$-cellulose | 10 | 10 | 10 | 10 | 10 |
| Bentonite | 10 | 10 | 10 | 10 | 10 |
| Ca(H$_2$PO$_4$)$_2$ | 20 | 20 | 20 | 20 | 20 |
| Vitamin and mineral premix * | 10 | 10 | 10 | 10 | 10 |
| Total | 1000 | 1000 | 1000 | 1000 | 1000 |
| Proximate composition (%) | | | | | |
| Crude protein | 39.2 ± 0.7 | 38.6 ± 0.7 | 39.4 ± 0.1 | 38.9 ± 2.7 | 38.5 ± 1.9 |
| Crude fat | 5.0 ± 0.6 | 7.3 ± 0.8 | 9.2 ± 0.7 | 11.1 ± 0.7 | 13.6 ± 0.3 |
| Ash | 9.6 ± 0.1 | 9.7 ± 0.1 | 9.8 ± 0.1 | 10.2 ± 0.1 | 9.7 ± 0.8 |
| Moisture | 11.3 ± 0.1 | 11.3 ± 0.3 | 10.7 ± 0.3 | 10.3 ± 0.6 | 10.3 ± 0.4 |

* The vitamin and mineral premix contained 10 g kg$^{-1}$ of vitamin mixture and 10 g kg$^{-1}$ of mineral. Each premix contained: Vitamin and minerals respectively contained: vitamin A 4000 IU (International Unit), vitamin D$_3$ 800 IU, vitamin E 50 IU, vitamin B$_1$ 2.5 mg, vitamin B$_2$ 9 mg, vitamin B$_6$ 10 mg, vitamin C 250 mg, nicotinic acid 40 mg, pantothenic acid calcium 30 mg, biotin 100μg, betaine 1000 mg, Fe 140 mg, Cu 2.5 mg, Zn 65 mg, Mn 19 mg, Mg 230 mg, Co 0.1 mg, I 0.25 mg, Se 0.2 mg.

*2.2. Sampling Procedures*

At the end of the feeding trial, all fish were sedated with tricaine methane sulfonate of water solution (MS-222; 0.01 g L$^{-1}$, Sigma-Aldrich, Burlington, MA, USA), and their BW and standard length (SL) were determined to the nearest 0.1 g and 1 mm, respectively. Once measured, blood samples were taken from the caudal peduncle vein with heparinized syringes from three fish per tank ($n$ = 9 per diet). The remaining fish from each tank were dissected for sex determination and to obtain the weight of selected tissues, such as the hepatopancreas, spleen, kidney, intestine, ovary, and testis. The gonads of ovary and testis of three fish per group were fixed, respectively, in 4% buffered formalin for 48 h for subsequent histological observation. The hepatopancreas and gonads were stored at −80 °C, respectively, for further biochemical analyses on fatty acid profile and gene expression studies.

*2.3. Growth Performance, Feed Utilization, and Somatic Indexes*

The effect of experimental diets differing in their lipid levels on the key performance indicators related to growth and feed performances in *O. macrolepis*, as well as on somatic parameters, were calculated using the following formulae:

Specific growth rate (SGR, %) = (Ln final BW − Ln initial BW) × 100/days;

Feed intake (FI, g fish$^{-1}$) = feed ingested (g)/number of fish;

Feed efficiency (FE) = BW gain (g)/feed intake (g);

Survival rate (SR, %) = (final number of fish/initial number of fish) × 100;

Viscerosomatic index (VSI, %) = weight of viscera (g)/BW (g) × 100;

Perivisceral fat index (PVFI, %) = weight of intraperitoneal fat (g)/BW (g) × 100;

Gonadosomatic index of female (GSIf, %) = weight of ovary (g)/BW (g) × 100.

Gonadosomatic index of male (GSIm, %) = weight of testis (g)/BW (g) × 100.

Hepatosomatic index (HSI, %) = weight of hepatopancreas (g)/BW (g) × 100;

Splenic somatic index (SSI, %) = weight of spleen (g)/BW (g) × 100;

Renal somatic index (RSI, %) = kidney weight (g)/BW (g) × 100;

Ratio of intestinal length to body length (RILBL, %) = length of intestine (cm)/SL (cm) × 100;

### 2.4. Proximate and Fatty Acid Composition of Diets and Fish Samples

Proximate composition of experimental diets, fish carcass, and tissues were analyzed according to the Association of Official Analytical Chemists (AOAC 2004) [27]. In particular, crude protein levels were determined by Kjeldahl's method, crude lipids were determined by the method of ethyl ether extraction, and moisture and ash contents were determined by sample drying in the oven at 105 °C for 24 h and by burning in a muffle furnace at 550 °C for four h, respectively.

For fatty acid analysis, lipids were extracted by chloroform/methanol (2:1 $v/v$) initiated by Folch et al. (1957) [28]; then, the lipid fraction was saponified and methylated using 12% boron trifluoride in methanol. Briefly, methyl esters were separated using a Gas chromatograph Trace-2000 (ThermoQuest Corporation, Austin, TX, USA) with the following procedure: 60 °C for 20 s, 25 °C min$^{-1}$; 160 °C for 28 min, 25 °C min$^{-1}$; 190 °C for 17 min, 25 °C min$^{-1}$; 220 °C for 9 min. This gas chromatograph was equipped with a 50-m CP-Sil 88 (Chromopack, Agilent) fused silica capillary column (i.d., 0.32 mm). Fatty acids were identified by retention time using standard mixtures of methyl esters (47015-U, Nu-Chek-Prep, Elysian, MN, USA) and quantified using Total Chrom software (version 6.2, Perkin Elmer) as described in Lie and Lambertsen (1991) [29]. The fatty acid composition of five diets was shown in Table 2.

**Table 2.** Fatty acid composition (%) of experimental diets differs in their total crude lipids.

| Fatty Acids | Experimental Diets | | | | |
|---|---|---|---|---|---|
| | 5 L | 7 L | 9 L | 11 L | 13 L |
| 14:0 | 3.43 ± 0.02 | 2.1 ± 0.01 | 1.59 ± 0.03 | 1.39 ± 0.03 | 1.04 ± 0.03 |
| 16:0 | 19.79 ± 0.14 | 15.7 ± 0.16 | 14.73 ± 0.17 | 14.38 ± 0.15 | 13.26 ± 0.19 |
| 18:0 | 3.79 ± 0.03 | 3.87 ± 0.04 | 4.02 ± 0.03 | 4.03 ± 0.07 | 4.04 ± 0.04 |
| 21:0 | 1.14 ± 0.03 | 0.19 ± 0.03 | 0.61 ± 0.03 | 0.54 ± 0.03 | 0.53 ± 0.02 |
| 24:0 | 4.26 ± 0.01 | 3.49 ± 0.02 | 2.55 ± 0.02 | 2.01 ± 0.11 | 1.70 ± 0.02 |
| ΣSFA | 32.41 ± 0.05 | 25.41 ± 0.05 | 23.50 ± 0.06 | 22.34 ± 0.08 | 20.57 ± 0.06 |
| 16:1$n$-7 | 3.24 ± 0.05 | 2.21 ± 0.06 | 1.70 ± 0.09 | 1.46 ± 0.05 | 1.19 ± 0.03 |
| 18:1$n$-9 | 25.94 ± 0.30 | 26.69 ± 0.31 | 26.61 ± 0.32 | 26.43 ± 0.15 | 26.3 ± 0.25 |
| ΣMONO | 29.18 ± 0.18 | 28.9 ± 0.19 | 28.31 ± 0.21 | 27.89 ± 0.10 | 27.49 ± 0.14 |
| 18:2$n$-6 | 21.09 ± 0.24 | 30.31 ± 0.20 | 34.64 ± 0.13 | 37.45 ± 0.07 | 39.79 ± 0.64 |
| 20:3$n$-6 | 0.15 ± 0.12 | 0.11 ± 0.01 | 0.11 ± 0.02 | 0.11 ± 0.03 | 0.13 ± 0.01 |
| 20:4$n$-6 | 0.25 ± 0.01 | 0.26 ± 0.01 | 0.23 ± 0.01 | 0.15 ± 0.03 | 0.15 ± 0.01 |
| Σ$n$-6 PUFA | 21.49 ± 0.12 | 30.68 ± 0.07 | 34.98 ± 0.05 | 37.71 ± 0.04 | 40.07 ± 0.22 |
| 18:3$n$-3 | 2.73 ± 0.03 | 4.21 ± 0.03 | 5.69 ± 0.02 | 6.07 ± 0.02 | 6.45 ± 0.22 |
| 20:3$n$-3 | 0.38 ± 0.01 | 0.36 ± 0.02 | 0.42 ± 0.01 | 0.25 ± 0.05 | 0.29 ± 0.05 |
| 20:5$n$-3 | 1.1 ± 0.02 | 0.34 ± 0.01 | 0.11 ± 0.04 | 0.03 ± 0.02 | 0.03 ± 0.12 |
| 22:6$n$-3 | 4.24 ± 0.03 | 4.10 ± 0.15 | 2.84 ± 0.11 | 2.22 ± 0.13 | 1.88 ± 0.35 |
| Σ$n$-3 PUFA | 8.45 ± 0.02 | 8.91 ± 0.05 | 9.06 ± 0.05 | 8.57 ± 0.06 | 8.65 ± 0.19 |

Abbreviations: SAT: Saturated fatty acids; MONO: Mono-unsaturated fatty acids; PUFA: poly-unsaturated fatty acids.

### 2.5. Histological Procedures

Histological procedures for gonad samples from different experimental groups were conducted according to the standard histological techniques [30]. In brief, samples were dehydrated in graded series of ethanol and embedded in paraffin. All tissue blocks were sectioned at the thickness of 5 μm and stained with Haematoxylin and Eosin (H & E), where gonads of ovary and testis were transected. All histological procedures were done at the Pathology Laboratory of the Yangling Demonstration Zone Hospital (Shaanxi, China). The morphology of the gonads were observed under a microscope (Motic BA310, Motic China Group Co., LTD manufacturer, Xiamen, China).

### 2.6. Gene Expression by Quantitative Real-Time PCR

The impact of dietary lipid level on protein metabolism and sex steroid synthesizing in selected tissues was assessed using the following selected genes: (i) protein

metabolism: insulin-like growth factor 1 (*igf1*), glutamate dehydrogenase (*gldh*); (ii) sex steroid-synthesizing: follicle stimulating hormone receptor (*fshr*), 3 beta hydroxysteroid dehydrogenase (*3β-hsd*), 17β-hydroxysteroid dehydrogenase (*17β-hsd*), aromatase (*aro.*), steroidogenic acute regulatory protein (*star.*).

Genes were determined by quantitative real-time PCR. Firstly, total RNA was extracted from the samples using RNAiso Plus (TaKaRa, Dalian, China) according to the manufacturer's instructions. Then, RNA was purified using the RNAWiz$^{TM}$ protocol (Ambion, Austin, TX, USA) to remove genomic DNA by DNase treatment (DNA-free$^{TM}$, Ambion), and its quality and quantity were determined by spectrophotometry (NanoDrop 1000, Thermo Scientific Inc., Wilmington, NC, USA) and agarose gel electrophoresis (2%). Then, RNA (1 μg) was reverse transcribed in a 20 μL reaction volume using random primers (Fermentas Life Science, Hanover, NH, USA); reverse transcription was carried out at 50 °C for 15 min and at 85 °C for 2 min, following the instructions of the kit manufacturer.

Gene expression was measured by quantitative real-time PCR with Sybr-Green (Roche, Switzerland), and reactions were performed using a CFX 96 Real-time PCR detection system (Bio-Rad, Hercules, CA, USA). The total volume of the PCR reaction was 20 μL, where 1 μL of the cDNA product, 0.6 μL of each primer (10 μM), 10 μL of 2 × SYBR$^{®}$ Premix Ex TaqTMII (TaKaRa, Dalian, China), and 7.8μL of double sterile distilled water. The RT-qPCR program started at 95 °C for 3 min, followed by 38 cycles (95 °C for 15 s, 60 °C for 30 s, and 65 °C for 5 s). The PCR primer sequences for each gene were synthesized by Sangon Biotech (Shanghai, China) Co., Ltd. and are shown in Table 3. Relative gene quantification was conducted according to the $2^{-\Delta\Delta CT}$ method [31], using *β-actin* as a housekeeping gene.

**Table 3.** List of primers used for quantitative PCR analysis.

| Gene | | Sequence (5′–3′) | Amplicon Size (bp) | PCR Efficiency (%) | Annealing Temperature (°C) |
|------|---|------------------|--------------------|--------------------|-----------------------------|
| *igf1* | F | CCACAGCCGGACCAGAGACC | 181 | 92 | 60 |
| | R | TCCAGCCTCCTCAGATCACAGC | | | |
| *gldh* | F | TGGCTTACACAATGGAGCGATCAG | 103 | 98 | 55 |
| | R | TTCTCAATGGCGTTGACGTAGGC | | | |
| *fshr* | F | ACCAGCATCTGCCTGCCAATG | 199 | 93 | 57 |
| | R | TGAAGATGAGCACGGCCATGC | | | |
| *3β-hsd* | F | TCGTTGCATGTGTCGGTGTGG | 128 | 94 | 58 |
| | R | GTGCAGGCCACAGCGAGTG | | | |
| *17β-hsd* | F | GCCTCTGTGGAAGGAGCTTGC | 109 | 95 | 58 |
| | R | TCCGCTCTCAGTCTCCGTTCC | | | |
| *aro.* | F | GCACCGTCAGCACCATCAAGC | 166 | 95 | 60 |
| | R | TGTGTCGCAGGCGGACTGG | | | |
| *star.* | F | GACCTGGACCTAGTGCCTGGATC | 148 | 94 | 57 |
| | R | GTGAGGATGCTGATGGACTTCTGC | | | |
| *β-actin* | F | GCCGGATTCGCTGGAGATGATG | 112 | 96 | 57 |
| | R | CACCAACGTAGCTGTCCTTCTGTC | | | |

Gene abbreviations: *igf1*: insulin-like growth factor 1; *Gldh1*: glutamate dehydrogenase 1; *fshr*: follicle-stimulating hormone receptor; *3β-hsd*: 3 beta-hydroxysteroid dehydrogenase; *17β-hsd*:17β-hydroxysteroid dehydrogenase; *aro.*: aromatase; *star.*: steroidogenic acute regulatory protein.

### 2.7. Statistical Analyses

Data are shown as means ± standard error. Data expressed as percentages were arcsine-square-root transformed before ANOVA analysis (data previously checked for normality and homoscedasticity). When significant differences were found ($p < 0.05$), the one-way ANOVA was followed by Duncan's post hoc test. All analyses were performed using SPSS 16 for Windows Software (SPSS, Chicago, IL, USA).

## 3. Results

### 3.1. Growth Performance and Somatic Indexes

At the end of the study, final body weight (BWf), specific growth rate (SGR), feed intake (FI), and feed efficiency ratio (FER) values in *O. macrolepis* brood fish were not significantly affected by dietary lipid levels ($p > 0.05$; Table 4).

**Table 4.** The effect of dietary crude lipid levels on growth and feed efficiency parameters of *Onychostoma macrolepis* broodfish.

| Index | Experimental Diets | | | | |
| --- | --- | --- | --- | --- | --- |
| | 5 L | 7 L | 9 L | 11 L | 13 L |
| BWi (g) | 48.95 ± 2.97 | 49.29 ± 3.16 | 51.71 ± 3.76 | 50.82 ± 3.10 | 49.76 ± 2.61 |
| BWf (g) | 56.11 ± 3.20 | 56.06 ± 5.29 | 61.14 ± 4.41 | 58.40 ± 3.52 | 57.56 ± 3.32 |
| SGR (% day$^{-1}$) | 0.24 ± 0.06 | 0.23 ± 0.06 | 0.30 ± 0.01 | 0.26 ± 0.00 | 0.26 ± 0.01 |
| FI (g) | 347.52 ± 5.02 | 337.62 ± 8.89 | 335.61 ± 3.22 | 335.92 ± 4.28 | 335.38 ± 12.37 |
| FER (%) | 14.69 ± 3.95 | 13.59 ± 3.59 | 18.25 ± 0.78 | 14.91 ± 0.11 | 15.65 ± 0.64 |
| SR (%) | 100 | 100 | 100 | 100 | 100 |

Abbreviations: BWi, initial body weight; BWf, final body weight; SGR, specific growth rate; FI, feed intake; FER, feed efficiency ratio; SR, Survival rate.

Regarding the impact of dietary lipid levels of somatic parameters on *O. macrolepis* females, GSIf, PVFI, and RSI were significantly affected by the diet ($p < 0.05$; Table 5). In particular, the highest GSIf values were found in broodfish from the 9 L (4.00 ± 0.83%) and 11 L (3.72 ± 0.45%) dietary groups, whereas the lowest values were those of females fed the 13 L diet (2.56 ± 0.12%). The rest of the groups (5 L and 7 L diets) showed intermediate values between maximal and minimal GSIf values. The highest and lowest levels of PVFI were found in females fed the 13 L (2.66 ± 0.44%) and 5 L (1.34 ± 0.09%) diets, respectively. The highest and lowest values in RSI were found in females fed the 13 L (2.67 ± 0.04%) and 5 L (1.34 ± 0.09%) diets, respectively. No statistically significant differences in the VSI, HSI, SSI, and RILBL were found between females fed different experimental diets ($p > 0.05$; Table 5).

**Table 5.** The effect of different dietary lipid levels on somatic indexes in *Onychostoma macrolepis* broodfish ($n = 3$; %).

| Items | Experimental Diets | | | | |
| --- | --- | --- | --- | --- | --- |
| | 5 L | 7 L | 9 L | 11 L | 13 L |
| Females | | | | | |
| VSI | 15.63 ± 2.49 | 15.64 ± 1.13 | 16.06 ± 2.75 | 16.24 ± 1.13 | 14.87 ± 1.56 |
| PVFI | 1.34 ± 0.09 [d] | 2.27 ± 0.41 [ab] | 2.09 ± 0.46 [bc] | 1.67 ± 0.52 [cd] | 2.66 ± 0.44 [a] |
| GSIf | 3.49 ± 0.36 [ab] | 3.17 ± 1.29 [ab] | 4.00 ± 0.83 [a] | 3.72 ± 0.45 [a] | 2.56 ± 0.12 [b] |
| HSI | 1.47 ± 0.12 | 1.31 ± 0.42 | 1.22 ± 0.23 | 1.35 ± 0.16 | 1.15 ± 0.24 |
| SSI | 0.18 ± 0.02 | 0.21 ± 0.07 | 0.17 ± 0.03 | 0.21 ± 0.08 | 0.22 ± 0.09 |
| RSI | 1.34 ± 0.09 [b] | 2.27 ± 0.04 [ab] | 2.09 ± 0.05 [ab] | 1.67 ± 0.52 [b] | 2.67 ± 0.04 [a] |
| RILBL | 252 ± 23 | 273 ± 19 | 614 ± 105 | 266 ± 22 | 277 ± 26 |
| Males | | | | | |
| VSI | 12.27 ± 1.58 | 12.65 ± 3.32 | 13.15 ± 2.44 | 12.89 ± 1.32 | 13.89 ± 1.44 |
| PVFI | 2.25 ± 0.35 | 2.55 ± 0.61 | 2.26 ± 0.45 | 2.54 ± 0.71 | 2.37 ± 0.72 |
| GSIm | 0.72 ± 0.17 [ab] | 0.67 ± 0.19 [b] | 1.33 ± 1.66 [a] | 0.70 ± 0.13 [ab] | 0.97 ± 0.81 [ab] |
| HSI | 1.32 ± 0.24 | 1.40 ± 0.30 | 1.23 ± 0.28 | 1.26 ± 0.27 | 1.22 ± 0.21 |
| SSI | 0.18 ± 0.03 [b] | 0.17 ± 0.05 [b] | 0.25 ± 0.08 [a] | 0.23 ± 0.06 [a] | 0.22 ± 0.06 [ab] |
| RSI | 0.55 ± 0.23 | 0.49 ± 0.12 | 0.55 ± 0.22 | 0.57 ± 0.23 | 0.49 ± 0.15 |
| RILBL | 250 ± 59 | 257 ± 22 | 260 ± 25 | 281 ± 50 | 257 ± 45 |

Abbreviations: VSI, viscerosomatic index; PVFI, perivisceral fat index; GSIf, gonadosomatic index of female fish; GSIm, gonadosomatic index of male fish; HSI, hepatosomatic index; SSI, splenic somatic index; RSI, renal somatic index; RILBL, ratio of intestinal length to body length. Different letters indicate statistically significant differences ($p < 0.05$).

Considering *O. macrolepis* males, different dietary lipid levels only affected GSIm and SSI values ($p < 0.05$; Table 5). In particular, the highest and lowest GSIm values were found in fish

fed 9 L ($1.33 \pm 1.66$%) and 7 L ($0.67 \pm 0.19$%) diets, respectively ($p < 0.05$; Table 5), whereas the rest of the dietary groups showed intermediate values. Regarding the SSI, the highest values were found in males fed 9 L ($0.25 \pm 0.08$%) and 11 L ($0.23 \pm 0.06$%) diets, whereas the lowest SSI values were recorded in males fed 5 L ($0.18 \pm 0.03$%) and 7 L ($0.17 \pm 0.05$%) diets ($p < 0.05$). The rest of the somatic indexes evaluated in males (HSI, VSI, PVFI, RSI, and RILBL) were not significantly different among dietary groups ($p > 0.05$; Table 5).

### 3.2. Fatty Acid Profile of the Gonad Tissues

The fatty acid composition of the gonad in *O. macrolepis* fed diets with different crude lipid levels was relatively stable in terms of total levels of saturated, monounsaturated, *n*-6, and *n*-3 polyunsaturated fatty acids (Table 6; $p > 0.05$). Statistically, significant differences were only found at the level of palmitic acid (16:0), linoleic acid (18:2*n*-6), and docosahexaenoic acid (22:6*n*-3) ($p < 0.05$). In particular, the highest and lowest levels of 16:0 were found in the testes of fish fed the 5 L and 7 L diets, respectively, whereas the rest of the dietary groups showed intermediate levels. Regarding 18:2*n*-6 levels, the highest values were found in the gonads of fish fed the 13 L diet, whereas the lowest values were found in fish fed the 5 L diet, while the rest of the dietary groups showed intermediate values. The content in 22:6*n*-3 was highest in the gonad of fish from the 5 L group, whereas the lowest values were found in fish from the 13 L, and the rest of the dietary groups showing intermediate values.

**Table 6.** Fatty acid composition (%) of gonad from *Onychostoma macrolepis* fed diets differing in their crude lipid content.

| Fatty Acids | Experimental Diets | | | | |
|---|---|---|---|---|---|
| | 5 L | 7 L | 9 L | 11 L | 13 L |
| 14:0 | $2.47 \pm 0.62$ | $1.78 \pm 0.36$ | $1.92 \pm 0.06$ | $1.59 \pm 0.08$ | $1.46 \pm 0.59$ |
| 16:0 | $17.33 \pm 2.14$ [a] | $13.20 \pm 0.66$ [b] | $16.03 \pm 0.76$ [ab] | $14.59 \pm 0.41$ [ab] | $14.80 \pm 0.54$ [ab] |
| 18:0 | $7.00 \pm 3.79$ | $5.15 \pm 2.03$ | $4.99 \pm 1.70$ | $3.73 \pm 0.58$ | $3.70 \pm 0.78$ |
| 21:0 | $2.55 \pm 0.60$ | $2.24 \pm 1.26$ | $1.23 \pm 0.11$ | $2.73 \pm 0.16$ | $2.96 \pm 0.01$ |
| 24:0 | $1.23 \pm 0.16$ | $1.88 \pm 0.25$ | $1.32 \pm 0.14$ | $1.45 \pm 0.08$ | $1.52 \pm 0.35$ |
| $\Sigma$SAT | $30.59 \pm 1.46$ | $24.25 \pm 0.91$ | $25.49 \pm 0.55$ | $24.08 \pm 0.26$ | $24.43 \pm 0.45$ |
| 16:1*n*-7 | $6.34 \pm 2.15$ | $4.62 \pm 1.06$ | $6.34 \pm 0.07$ | $4.52 \pm 0.17$ | $5.67 \pm 0.62$ |
| 18:1*n*-9 | $21.21 \pm 9.80$ | $18.93 \pm 2.78$ | $26.37 \pm 1.70$ | $26.92 \pm 1.96$ | $30.80 \pm 0.73$ |
| $\Sigma$MONO | $27.54 \pm 5.98$ | $23.56 \pm 1.92$ | $32.71 \pm 0.89$ | $31.44 \pm 1.07$ | $36.47 \pm 0.68$ |
| 18:2*n*-6 | $8.85 \pm 3.91$ [b] | $14.10 \pm 2.04$ [ab] | $14.91 \pm 8.24$ [ab] | $22.83 \pm 3.20$ [a] | $21.86 \pm 3.22$ [a] |
| 20:3*n*-6 | $1.82 \pm 0.79$ | $2.36 \pm 1.21$ | $1.50 \pm 0.66$ | $1.22 \pm 0.28$ | $1.16 \pm 0.08$ |
| 20:4*n*-6 | $0.33 \pm 0.16$ | $0.41 \pm 0.27$ | $0.11 \pm 0.00$ | $0.08 \pm 0.02$ | $0.07 \pm 0.05$ |
| $\Sigma$*n*-6 PUFA | $11.00 \pm 1.62$ | $16.86 \pm 1.17$ | $16.52 \pm 2.97$ | $24.12 \pm 1.17$ | $23.08 \pm 1.12$ |
| 18:3*n*-3 | $1.97 \pm 0.23$ | $2.04 \pm 1.01$ | $2.64 \pm 0.60$ | $3.48 \pm 0.47$ | $3.09 \pm 0.43$ |
| 20:3*n*-3 | $3.47 \pm 2.34$ | $2.80 \pm 1.01$ | $1.82 \pm 0.36$ | $1.27 \pm 0.08$ | $1.50 \pm 0.23$ |
| 20:5*n*-3 | $1.28 \pm 0.42$ | $1.01 \pm 0.77$ | $0.90 \pm 0.12$ | $0.87 \pm 0.05$ | $0.40 \pm 0.32$ |
| 22:6*n*-3 | $12.18 \pm 0.72$ [a] | $7.88 \pm 1.36$ [b] | $5.54 \pm 1.08$ [bc] | $4.76 \pm 0.84$ [bc] | $3.94 \pm 0.83$ [c] |
| $\Sigma$*n*-3 PUFA | $18.91 \pm 0.93$ | $13.73 \pm 1.04$ | $10.88 \pm 0.54$ | $10.37 \pm 0.36$ | $8.92 \pm 0.45$ |
| LNA/LA | $0.22 \pm 0.06$ | $0.14 \pm 0.09$ | $0.17 \pm 0.07$ | $0.15 \pm 0.1$ | $0.14 \pm 0.1$ |

Abbreviations: SAT: Saturated fatty acids; MONO: Monounsaturated fatty acids; PUFA: Polyunsaturated fatty acids. Data were obtained from six different specimens per diet. Values with different superscripts in each row are significantly different ($p < 0.05$).

### 3.3. Histological Organization of Target Tissues

The cell morphology and structural integrity of cell membranes of the testis and ovary in fish of five dietary groups were exhibited as clear and intact, and no abnormal cells were found (Figure 1A). The ovaries are composed of oocytes at the different developmental stages, including some more giant mature cells, and appeared granulated due to the high concentration of yolk globules, as well as some smaller under mature cells. The mean egg diameter showed no significant difference in different diets, and oocyte atresia was not noted in the ovaries of females among these experimental groups (Figure 1B, data not shown fully).

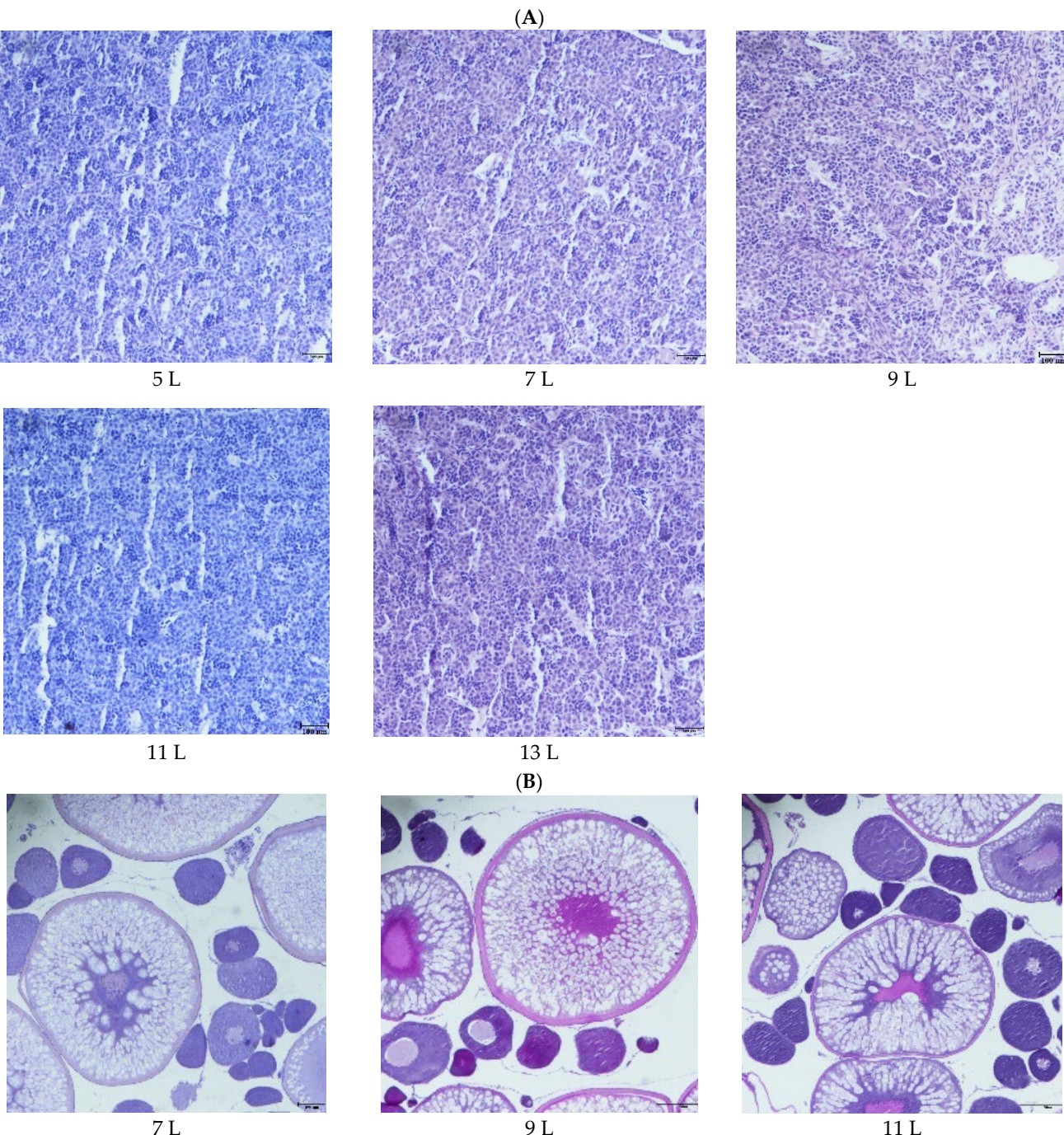

**Figure 1.** Histological section of the testis and ovary of the male and female *O. macrolepis* brood-stock, fed graded lipid levels during 8 weeks of feeding trial. (Testis, 100× magnification; ovary, 40× magnification; bar = 100 μm). Here only show histological section of the ovary in 7 L, 9 L and 11 L groups. (**A**). Gonad of male (testis); (**B**). Gonad of female (overy).

### 3.4. Gene Expression

The relative expression of genes involved in growth and protein metabolism in the hepatopancreas of *O. macrolepis* fed diets with different lipid levels are illustrated in Figure 2A. In particular, relative gene expression values for *igf* and *gldh* were similar among other experimental groups (Figure 2A).

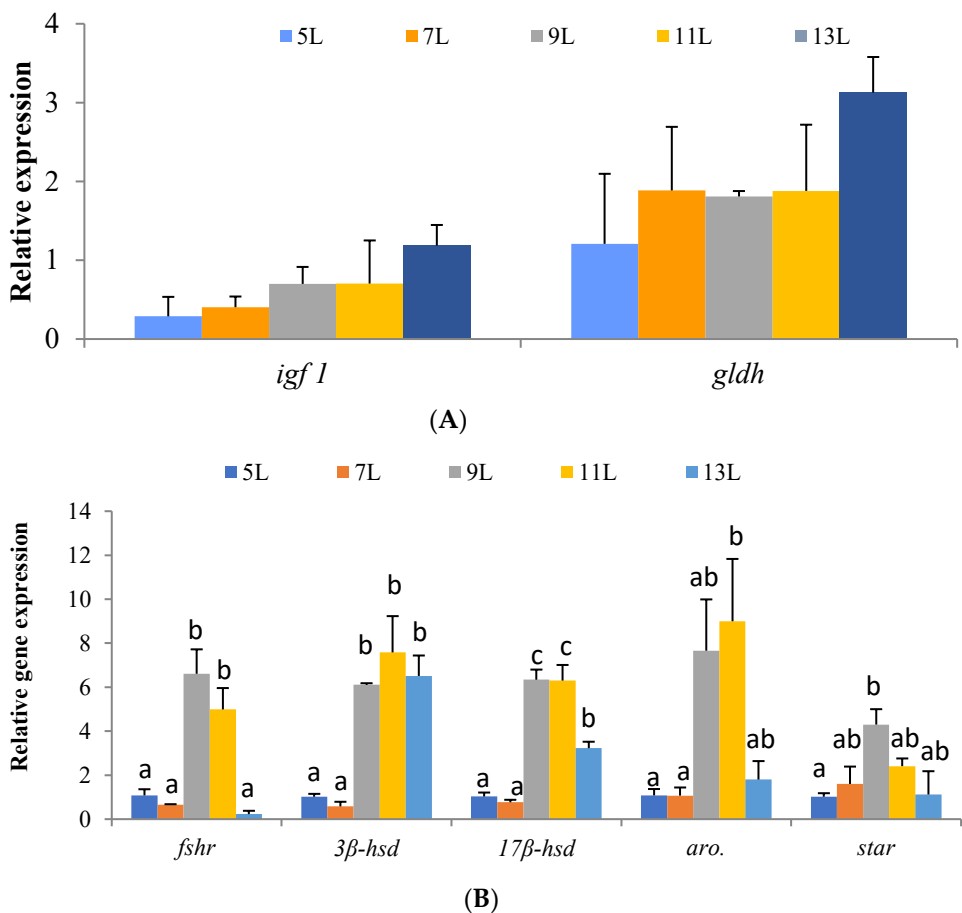

**Figure 2.** Relative expression values of genes from the hepatopancreas involved in growth and protein metabolism (*igf1*, *gldh*; (**A**)) and the gonad involved in sex steroid-synthesizing proteins (*fshr*, *3β-hsd*, *17β-hsd*, *aro.* and *star*.; (**B**)) in *Onychostoma macrolepis* broodfish fed diets differing in their crude lipid levels. Different letters indicate significant differences.

Gene expression of markers involved in gonad development is shown in Figure 2B. In particular, we found that dietary lipid levels modulated the expression of *fshr*, *3β-hsd*, *17β-hsd*, aro, and star in the gonad of *O. macrolepis* ($p < 0.05$). Maximal relative expression values of *fshr* were found in fish fed 9 L and 11 L diets, whereas the rest of the dietary groups showed similar expression levels. Regarding *3β-hsd*, the highest expression values were found in fish fed the highest dietary lipid levels (9 L, 11 L, and 13 L diets), whereas the lowest gene expression levels were found between fish fed 5 L and 7 L diets. The expression levels of *17β-hsd* were maximal in testes of fish fed 9 L and 11 L diets, whereas the lowest ones were in testes from the 5 L and 7 L groups. In contrast, testes from animals fed the 13 L diet showed intermediate expression levels.

## 4. Discussion

### 4.1. Effect of Lipid Level on Growth of O. macrolepis Broodstock

Many previous results usually showed that higher lipid levels induced higher growth of broodstock fish, such as Snakehead murrel [14] and European sea bass [32], because dietary protein can be readily used as a source of energy [33], and the catabolism of dietary lipid over protein can help spare the use of this costly nutrient, demonstrating the protein-sparing effect of lipid. In contrast, high dietary lipid levels did not inevitably induce higher growth of fish, which had also been observed in many fish, such as large yellow croaker [34], hybrid snakehead fingerling [35], Senegalese sole juveniles [36], giant croaker [37], and hybrid yellow catfish [38], which was in agreement with the present study that high dietary lipid levels did not induce higher growth of *O. macrolepis* broodstock fish (Table 4).

To further explore the growth of *O. macrolepis* broodstock fish, two genes growth-related properties were determined. The *ifg1*, being a growth hormone/insulin-like growth factor (IGF), regulates growth and cellular metabolism [39] and plays essential role in the growth and development of many fish, e.g., *Micropterus salmoides* [39], *Oncorhynchus kisutch* [40], and *Lateolabrax japonicus* [41], and *ifg1* gene expression was generally affected by nutritional status in fish [42,43]. The *gldh*, glutamate dehydrogenase (GDH), reversibly catalyzes glutamate deamination with the production of ammonia and plays a key role in nitrogen metabolism and the production of plasma ammonia [44,45]. In the present study, an increased trend of *ifg1*, *gldh* was observed in *O. macrolepis* broodstock fish fed diets with lipid levels from 5% to 13%, which was in agreement with the growth of fish. The present non-positive response of growth of *O. macrolepis* broodstock fish to the high dietary lipid intake is probably due to the slow growth characteristics of *O. macrolepis*, being a small-sized cave fish and a sub-cold-water fish [46]. In fact, the mean weight of 2–3-year-old adult female and male *O. macrolepis* is less than $38.79 \pm 0.80$ g and $17.56 \pm 0.33$ g, respectively, in the natural environment [26], still being tiny size fish, which indicates the growth rate inevitably is relatively low. This low and slow growth feeding with high dietary lipid levels had also been reported in white seabream [47] and hybrid snakehead fish [35].

### 4.2. Effect of Lipid Level on Gonad Development of O. macrolepis Broodstock

Optimizing lipid in a diet during feed formulation promotes the growth of fish. A conflict between somatic and gonadal growth, especially under restricted food conditions, has often been discussed [48], while in the present study, the *O. macrolepis* broodstock was fed under the condition of sufficient nutrition supply, then the somatic and gonadal growth were sufficiently supported by diet nutrients.

Gonadosomatic indexes (GSI, typically expressed as $100 \times$ gonad mass/total body mass) show the proportion of body tissue devoted to gamete production, and can be an appropriate proxy for gonadal development and gonad expenditure [49]. The mass increase in GSI is attributable to the growth of nutritive phagocytes that accumulate nutrients of protein and lipids derived from ingested food [50]. GSI, being an important indicator to evaluate fish gonadal development, provides a quantitative basis for assessing fish gonadal development potential and was generally affected by many factors, such as photothermal [51] influence, and nutrients, such as vitamin A, C, and E [3–6], protein and carbohydrate levels [7], and lipid levels [14–16]. In the present study, the gonad indexes of female (GSIf) in *O. macrolepis* broodstock increased with increasing dietary lipid levels from 5% to 11%, and they reached the highest in group 9 L–11 L, being 3.72–4%, while with further increasing of lipid to 13%, it decreased ($p < 0.05$; Table 5). This result was a little different from previous findings in snakehead murrel Channa striatus, where GSIf of snakehead murrel increased with dietary lipid level (100–180 g $kg^{-1}$; 10–18%) [13]. This difference probably was caused by the fact that *O. macrolepis* is an omnivorous and phytophagous fish; the lipid level of 13% (130 g $kg^{-1}$) probably is too high and excessive for its ovary growth and development; while snakehead murrel is a carnivorous fish [52], and it requires higher dietary lipid (e.g., 180 g $kg^{-1}$) for gonad development.

Between male and female fish, previous studies paid less attention to gonadosomatic indexes of males (GSIm) compared to that of females [14,16], and in male fish, researchers paid more interest to sperm motility and sperm freeze–thaw fertilizing ability [53]. Different polyunsaturated fatty acids affecting sperm fertilizing ability were reported in fish. Eicosapentaenoic (EPA) and arachidonic acid (ARA) levels exerted a marked effect on fertilization in species such as rainbow trout [54,55] and European seabass [56] by affecting their sperm motility, while no effect of dietary fatty acids *n*-3 and *n*-6 polyunsaturated fatty acids on sperm freeze–thaw fertilizing ability was found by Labbe et al. [55]. Although the present result did not provide data on sperm motility or sperm fertilizing ability, the GSIm showed less correlation with dietary lipid levels ($R^2 = 0.1898$) compared with GSIf ($R^2 = 0.5965$), probably indicating that gonad development of male *O. macrolepis* broodstock is less affected by dietary lipid levels, and the morphology and membrane integrity of sperm cells



were not found to be different among dietary lipid groups (Figure 1), which probably provided more evidence for these findings.

The previous results in *O. macrolepis* broodstock showed that the oocytes were increasingly more prominent and more mature with the increase of dietary protein levels [26]. The present research is the first to study the effect of dietary lipid levels on gonad development of *O. macrolepis* broodstock. The current results showed that no significant differences were noted in the histological features of ovary sectioning in *O. macrolepis* broodstock among five dietary groups; a similar result was found in orange mud crabs' ovaries [57]. While in Chinese sturgeon, higher dietary lipid levels improved ovarian development, with a higher percent of mature oocytes and larger giant oocyte (diameter), especially for the 24-month feeding period [58]. The difference between the result of *O. macrolepis* and Chinese sturgeon probably is caused by the different fish species, feeding periods, and specific lipid levels used.

Previously the use of hormone injections for spermiation and reproduction of *O. macrolepis* broodstock had been reported [59], while the research about the histology of *O. macrolepis* broodstock testes during maturation generally is scarcely concerned about consulting the same kinds of literature, or about the effect of dietary lipid levels on the histology of *O. macrolepis* broodstock testis, which is firstly reported in the present study. No significant differences in the histological features of testis sectioning were observed in *O. macrolepis* broodstock among the five dietary groups.

Sex-steroid hormones mainly include testosterone and estradiol. Testosterone can promote spermatogenesis, promote the development and differentiation of the reproductive duct, and maintain reproductive function. Estradiol can promote the growth and maturation of follicular cells and directly affect the secretory function of the ovary. They both can stimulate gonad maturation and function, and their level can reflect the maturity degree of the gonad, being a vital index to accurately judge the reproductive state of broodfish [51]. Conversion of cholesterol to sex-steroid hormones in gonad cells is a complex progress mainly regulated by the steroid metabolic pathway [60]. In this pathway, the genes of *fshr*, hsd, aromatase (*aro.*), and steroidogenic acute regulatory protein (star) were taken as the markers for steroidogenesis in gonad development and maturation [61–65]. Star is a hormone-induced mitochondria-targeted protein that has been shown to initiate cholesterol transfer into mitochondria, which is considered a rate-limiting step in hormone-induced steroid formation [64].

By feeding experiment, the present investigation studied the effect of lipid levels in feed on fish steroid hormone synthesis, including the critical process of steroid hormone synthesis (i.e., response to gonadotropin (*fshr*), cholesterol transport (*star.*), and enzymatic reaction process (*3β-hsd*, *17β-hsd*, *aro.*), respectively, by molecular biological methods, to reveal the relevant mechanism and process of LL regulating steroid hormone synthesis. These fundamental processes are the core processes of steroid hormone synthesis, but there are few studies on their regulation by feed nutrients [66]. The present results showed that most of these genes were highly expressed in groups 9 L and 11 L (Figure 2), indicating that proper dietary lipid level is suitable for steroidogenesis and gonad maturation for *O. macrolepis* broodfish.

With the deepening of study, the effect of dietary fatty acids are more substantially concentrated than lipid levels on the reproductive performance of fish [1,67–71], especially when one of the polyunsaturated fatty acids, arachidonic acid (ARA), is found to be the precursor of eicosanoids and has been proven to influence sex steroidogenesis [65,72]. Although Zhang et al. (2017) [66] found that ARA inhibited the estradiol and testosterone production in immature turbot Scophthalmus maximus broodstock, Norambuena et al. (2013) [73] found that ARA in feeds could increase the content of 11 keto testosterone and testosterone in the blood of fish, and the latter finding was supported by cell experiments in vitro in goldfish (*Carassius auratus*) [74–76]. Although the content of the sex hormone in the blood of *O. macrolepis* broodfish is unknown in the present study, the ARA content of the gonads was not significantly affected by dietary lipid levels ($p > 0.05$; Table 5).

More proof is required to certify the relationship between the sex hormone in the blood of fish and the dietary lipid level, while the relative expression values of genes involved in sex steroid-synthesizing proteins speaks for itself.

## 5. Conclusions

In summary, the present data revealed that these five dietary lipid levels did not affect the growth of *O. macrolepis* broodstock, while the dietary lipid levels of 90–110 g kg$^{-1}$ promoted higher gonad development with considerable higher gonadosomatic indexes and relative expression of steroid hormone synthesis related genes, suggesting that a proper dietary lipid level of 90–110 g Kg$^{-1}$ could maintain the growth performance and gonad development of *O. macrolepis* broodstock. The same growth of broodstock in five lipid levels probably is related to the fact that it is a relatively small-sized and naturally slow-growing cave fish. The GIS, especially the GSIf, firstly increased with the increasing dietary lipid level and then decreased with the further increasing dietary lipid level, probably indicating that dietary lipid was mainly used in the growth of the gonad. This is an interesting finding, and further research is required to explore this phenomenon.

**Author Contributions:** Conceptualization, H.J.; methodology, Y.L.; validation, P.F.; formal analysis, J.Z.; writing—original draft preparation, J.Z.; writing—review and editing, all authors; visualization, E.G.; supervision, J.Z.; project administration, J.Z.; funding acquisition, H.J. All authors have read and agreed to the published version of the manuscript.

**Funding:** This study was financially supported by the National Key R & D Program of China (2019YFD0900200), Special R & D Program Project of Chinese Academy of Se-enriched Industry (2019ZKG-1), Key R & D plan of Shaanxi Province (2018ZDXM-NY-045) and Shaanxi Special plan project of technological innovation guidance (2022QFY12-03).

**Institutional Review Board Statement:** In This study was conducted following the Guiding Principles for Biomedical Research Involving Animals (EU2010/63), the guidelines of the Spanish laws (law 32/2007 and RD 53/2013) and authorized by the Ethical Committee of IRTA (Spain) and by the Ethical committee of NWAFU (China) for the use of laboratory animals.

**Informed Consent Statement:** Not applicable.

**Data Availability Statement:** The data that support the findings are available upon direct request to the corresponding author.

**Acknowledgments:** The authors would like to acknowledge Wang Gang for his kind donation of fish used in this study and to Liu Hai Xia for the provision of her result of transcriptome in *O. macrolepis* being helpful to design the primers of these genes tested in the present paper.

**Conflicts of Interest:** The authors declare no conflict of interest.

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
