# Peer review of "Effects of Dietary Lipid Levels on Growth and Gonad Development of Onychostoma macrolepis Broodfish"

_fishes, doi:10.3390/fishes7050291_

Round 1

Reviewer 1 Report

Title: Effects of dietary lipid levels on Onychostoma macrolepis broodfish

Some points are important to be addressed before going to accept this article.   

1.     Keywords also need to improve with some related keywords

2.     The introduction should improve with some recent publications, especially in the fatty acids profiles for the lipid-sources ingredients.

3.     In all the manuscript, please, make a space like this (P > 0.05).  As well as, the (P > 0.05) is only added after the word "significant difference"

4.     In all the manuscript, revise all scientific names and be sure it was Italic.

5.     The conclusion should improve

Author Response

Response to Reviewer 1 Comments

Point 1:  1.     Keywords also need to improve with some related keywords

Response 1:  Yes, the number of keywords increased from “aquaculture; lipid; broodstock” to “aquaculture; lipid; broodstock; growth; gonad development”. Thank you for your suggestion.

Point 2:  2.     The introduction should improve with some recent publications, especially in the fatty acids profiles for the lipid-sources ingredients.

Response 2: Yes, lipid sources is different in the fatty acids profiles which will induce the fish’s fatty acids profiles is different because of the characters of lipid uptake and metabolism. In the present experiment it concentrates on the effect of lipid levels but not lipid sources on the fish. In the present study different levels of the soybean oil was added in the diets to form five lipid level diets and the dietary fatty acids profiles is supplemented in the part of material and methods in the new version. Thank you for your suggestions.

Point 3:  3.     In all the manuscript, please, make a space like this (P > 0.05).  As well as, the (P > 0.05) is only added after the word "significant difference"

Response 3: Yes, it is vertivised to have a space in (P > 0.05) and the (P > 0.05) is only added after the word "significant difference". Thank you for your suggestions.

Point 4:  4.     In all the manuscript, revise all scientific names and be sure it was Italic.

Response 4: Yes, all scientific names is vertivised to be italic. Thank you for your suggestions.

Point 5:  5.     The conclusion should improve

Response 5: Yes, the conclusion is verifised and improved in a certain extent. Thank you for your suggestions.

Reviewer 2 Report

The paper has scientific merit. However there are several issues need to revise before can be published. The issues are as follows:

Title - accepted

Abstract -  well elaborate

Introduction

If possible to provide market value and market demand of the aquaculture species

Materials and methods

Please provide each references for each subtitle to justify methods and materials used in the study

Results & discussion - accepted

Conclusion

Please put more effort on this section. Provide research gap, future work and etc 

Author Response

Response to Reviewer 2 Comments

Point 1: Introduction 

If possible to provide market value and market demand of the aquaculture species

 Response 1: Yes, the market value and nutritious value is provided in the revised version with the price of being about 200 RMB per kg. Thank you for your suggestion.

Point 2: Materials and methods

Please provide each references for each subtitle to justify methods and materials used in the study

Response 2: Yes, in the verified version, more references is provide for each methods, such as the methods for proximate composition by AOAC and fatty acid composition by gas chromatography. Thank you for your suggestion.

Point 3: Results & discussion - accepted

Response 3: Thanks.

Point 4: Conclusion

 Please put more effort on this section. Provide research gap, future work and etc 

Response 4: Yes, some future work had been brought out in the part of conclusion. Thank you for your suggestion.

Author Response

Response to Reviewer 3 Comments

Explanation before the Response to Reviewer 3 Comments. As the reviewer 3 give comments in the form of PDF, then the points of his/hers is taken as followed and the response is given one by one. 

Point 1: The author have done a lot of work however following fed points are necessary to address and in its present form it can not be recommended for publication. Following are the points and a through revision of languages , phrases is needed.

Response 1: I am listening with open ears.

Point 2: Title: it looks very ordinary, it needs to be changed.

Response 2: The title is revised as “Effects of dietary lipid levels on growth and gonad development of Onychostoma macrolepis broodfish”, where the words of “growth and gonad development” is newly supplemented. The new title is more specific. Thank you for your suggestion.

Point 3: Abstract: line 12-13 , not mentioned CP %.

Response 3: Yes, the dietary CP is 39% and it showed in table 1 and it is supplemented in the abstract of the new versione. Thank you for your reminding.

Point 4: Introduction: Line 34-37 needs to be rewritten. Sentence should be clear and short for better understanding. Insert recent reference where ever applicable.

Response 4: Line 34-37 is attached followed: “Given the increasing importance of domestication in aquaculture, there is a need for an increased focus on the role of nutrition in broodstock and gonad maturation, as well as in improving larval quality and performance. Izquierdo et al. (2001) [1] pointed out that broodstock nutrition of fin fish nutrition was one of the most poorly understood and re-searched areas almost two decades ago, and significant advances have been achieved in this area [2] in these years, there are still important gaps of knowledge related to fish broodstock nutrition.”. These two sentences are long, while they reads smoothly. The logical meaning of the sentence is very clear. There is no mistakes in grammar too. Thank you for your stuggestion, while when I try to make it shorter I failed. 

About your suggestions of inserting some recent reference in the introduction, yes, some recent reference published in about 2013, 2014, 2015 and even 2018, were found and then they are supplemented in the introduction where applicable. Thank you for your suggestion.

Point 5:  2. Materials and Methods

Can you provide ethical approval number if possible? Line 71: Scientific name to be italized. Which experimental design you have used not mentioned any thing and no control ten how will compare the effects without control.

Response 5: Yes the ethical approval number of our college is DK2022066.

Yes, all scientific names, including O. macrolepis, are italized in the new revised version.

There are about three models to estimate quantitative nutrients requiremnts, including broken-line model, nonlinear models and factorial model, which have been described in the book of NRC “Nutrient requirement of fish and shrimp” in detail, pp11-13. It read in the book that : “Requiremnt estimates are typically based on experiments with regression design”. Pp 9.

Point 6:  In table 1 mention about the gross emery value.

Response 6: According to the tables of feed composition and nutritive value in china (2019), the total energy of feed gross energy is calculated by the following formula:

Total energy of feed GE (MJ/kg DM)=(4153+(56 * EE)+(15 * CP) - 44 * Ash) * 0.0041868.

This calculation formula was brought out by Ewan (1989), which is obviously very complicated, where crude fat, crude protein and even ash of each feed sources should be given before calculating.

Another good index to show the energy of each diets is “the effective energy” and the “tables of feed composition and nutritive value in china” gives the effective energy value of each feed sources, like fish meal, soybean meal and etc.., while the effective energy is mainly from the pig, chicken, cow and goat, so no effective energy can be refered from fish. It’s difficult to get the effective energy of a specific fish for the complicated water temperature, feeding condition and feeds used. If it is very necessary to get the energy, it‘s better to refer from the effective energy of pigs.

Point 7:  In table 1 see the trend and moisture logical its not fir so I advise to recheck the values.

Response 7: The five diets are formulated by feed resources and dried in the same condition, so they should have the same moisture level. The difference of the diets are the lipid levels, with the certain variation of carbohydrates, where it dosen’t show in the table 1 for it can be calculated by the content of protein, lipid, and even moisture.

Point 8:  2.2. sampling procedure Recheck the dose of MS222, is it ok??

Response 8: In previous experiment the dose of MS222 solution used was 0.1 g L-1 to sedate the Atlantic salmon (Zhou et al., 2014), while in the present experiment the dose of MS222 solution was 0.01 g L-1 considering the O. macrolepis is smaller than the Atlantic salmon. The reference of Zhou et al.,(2014) is followed:

Zhou Ji-Shu, Bente E. Torstensen and Ingunn Stubhaug. Oleic acid trans-membrane uptake in hepatocytes of Atlantic salmon (Salmo salar L.) and effect of replacing dietary fish oil with vegetable oil. ACTA HYDROBIOLOGICA SINICA, 2014, 38:137-144. (in Chinese)

Point 9:  2.3. Growth performance, feed utilization, and somatic indexes Add WG %, initial value of weight, ADG, PER, FCR, and if possible Temperature Growth coefficient.

Response 9: Yes, in this study, BWi (initial body weight), BWf (final body weight), SGR (specific growth rate), FI (feed intake), FER (feed efficiency ratio) and SR (Survival rate) were determined respectively. This is the typical indexes to show the effect of feeds on growth of fish. ADG is the apparent digestive energy, which require the collection of feces during the feeding period, while as the feces is easy to disolve in the water and no more effective methods is found to collect the feces, the feces is not collected in the study. Temperature Growth coefficient is a good index to show the growth of fish, which requires close detection of water temperature, while in the present study, the water temperature was only read as a reference to feed the fish or a judgement to observe the feeding activity, the frequency of detecting the water temperature is about once a day or twice a day, which is not met with the requirement of Temperature Growth coefficient, so this index is not detected in this study. Anyhow, this is a good suggestion and in the next feeding experiment this index would be considered. Thank you for your suggestion.

Point 10:  Table 2. List of primers used for quantitative PCR analysis. Why the β-actin amplicon size is too small it is a house keeping gene so approx. 200 bps size you should have used.

Response 10: The primers size of β-actin amplicon used for qPCR is small, which had been used previously in O. macrolepis or in other fish species.

The “Table 1” followed is taken from an article:”Yu et al. 2017. Antioxidant defenses of Onychostoma macrolepis in response to thermal stress: Insight from mRNA expression and activity of superoxide dismutase and catalase. Fish & Shellfish Immunology 66, 50-61”, where it shows that β-actin amplicon size is 20 bps.

The “Table 3” followed is taken from an article:”Ma et al., 2020. Effect of dietary linolenic/linoleic acid ratios on growth performance, ovarian steroidogenesis, plasma sex steroid hormone, and tissue fatty acid accumulation in juvenile common carp, Cyprinus carpio. Aquaculture Reports 18 , 100452”, where it shows that β-actin amplicon size is from 20 to 23 bps.

Therefore the primer of house keeping gene used for qPCR should be small. Anyhow, thank you for your suggestion and in our future research this 200bps primers for this gene would be considered seriously.

I have tried to attach the tables in the letter, while it failed. Anyhow the reference showing the primer size of actin is there and it's easy to find it. 

Point 11:  Table 3. The effect of dietary crude lipid levels on growth and feed efficiency parameters of Onychostoma macrolepis broodfish Showed that dietary lipid does not have any significant effects on different level is it true. I don’t think so 5-13 range is to vast as far as the lipid level is concerned, and what is the optimum lipid requirement of the species.

Response 11: Yes, as it was discussed in the part of discussion, this fish is a kind of small and growth-slow cave fish, where it is about 20-39 g in 3 years old fish. In the present study, the mean weight in the initial and final feeding period is about 49 g and 57 g respectively, which certify the slow growth of fish. While the gonadosomatic index, especially the gonadosomatic index of female fish firstly increased with the increasing dietary lipid level and then decreased with the further increasing dietary lipid level, probably indicating that dietary lipid are mainly used in the growth of the gonad of fish but not the growth of muscle. This is an interesting phenomenon. Considrring the growth of gonadosomatic index of fish, 90-130 g per kg lipid level is suggested in the present study. Thank you for your suggestion.

Point 12:  Line ;199 Scientific name to be italicized, it seems author are in hurry to submit the article.

Response 12: The Line ;199 Scientific name is italicized and all scientific names in the whole ms are italicized by careful check-up. Thank you for your suggestion.

Point 13:  The pic of gonald of male fish is not clear provide more clear picture.

Response 13: The picture of gonald of male fish was taken concurrently with that of the female fish and it was finished by the microscope in a key lab. Obviously the cells in the gonad of female fish is less, being clear, while the cells in the gonad of male fish is huge, which do not make it looks clear. So the picture of gonad in the male fish truly reflect the spermatocytes. Thank you for your suggestion.

Point 14:  3.3. Histological organization of target tissues In material and methods section it was written that author have taken intestine also for histological studies but no data have been provided.

Response 14: Yes, initially the histological organization of target tissues include hepatopancreas, intestine, then it is abandoned for they don’t looks different and this ms would be too long to include this part, so they are not provied in the result. The writing in the material and methods section has been verified. Thank you for your suggestion.

Point 15:  3.4. Gene expression Not provided the expression level of housekeeping gene then how you compared the fold change value in different treatments.

Response 15: In material and methods section relative gene quantification was conducted according to the 2 (-Delta Delta C(T)) method, where the author of this reference is Livak & Schmittgen, which is followed. By this method, the expression of housekeeping gene is subtracted twice to get the relative expression of target genes, so the relative expression level of housekeeping gene would not been given for it is object of reference.

Livak, K.J.; Schmittgen, T.D. Analysis of relative gene expression data using real-time quantitative PCR and the 2△△Ct method. Methods 2001, 25, 402-408.

Point 16: 4.2. Effect of lipid level on gonad development of O.macrolepis broodstock Check the abbreviation for gonadosomatic index whither it is GI and GSI or only GSO or both

Response 16: Yes, there are different forms to express this indexes, being gonad index (GI) or gonadosomatic index (GSI) in different reference. The referce with GI and GSI is followed.

The reference to use GI:

  1. Cuesta-Gomez, D.M.; Sánchez-Saavedra, M. D. P. Effects of protein and carbohydrate levels on survival, consumption and gonad index in adult sea urchin Strongylocentrotus purpuratus (Stimpson 1857) from Baja California, Mexico. Aquac Res 2017, 48, 1596-1607.

The reference to use GSI:

  1. Parker, G. A.; Ramm, S. A.; Lehtonen, J.; Henshaw, J. M. The evolution of gonad expenditure and gonadosomatic index (GSI) in male and female broadcast-spawning invertebrates. 2017. https://doi.org/10.1111/brv.12363.
  2. Ghaedi, A.; Kabir, M.A.; Hashim, R. Effect of lipid levels on the reproductive performance of Snakehead murrel, Channastriatus. Aquac Res 2016, 47, 983-991.

Round 2

Reviewer 3 Report

The author has addressed all the issues raised so now it can be accepted for publication.